# Isolation and Characterization of Biocontrol Microbes for Development of Effective Microbial Consortia for Managing *Rhizoctonia bataticola* Root Rot of Cluster Bean Under Hot Arid Climatic Conditions

**DOI:** 10.3390/microorganisms12112331

**Published:** 2024-11-15

**Authors:** Devendra Singh, Neelam Geat, Kuldeep Singh Jadon, Aman Verma, Rajneesh Sharma, Laxman Singh Rajput, Hans Raj Mahla, Rajesh Kumar Kakani

**Affiliations:** 1Division of Plant Improvement and Pest Management, ICAR-Central Arid Zone Research Institute, Jodhpur 342003, India; kuldeep.icar@gmail.com (K.S.J.); verma.aman1980@gmail.com (A.V.); rajneesh.sharma@icar.gov.in (R.S.); laxman0742@gmail.com (L.S.R.); hans.mahla@icar.gov.in (H.R.M.); rajesh.kakani@icar.gov.in (R.K.K.); 2Department of Plant Pathology, Agricultural Research Station, Mandor, Agriculture University, Jodhpur 342304, India; nilugeat@gmail.com

**Keywords:** *Rhizoctonia bataticola*, biocontrol, plant growth-promoting activities, consortium, systemic resistance, antioxidant defense enzyme

## Abstract

Development of native microbial consortia is crucial for the sustainable management of plant diseases in modern agriculture. This study aimed to evaluate the antagonistic potential of various microbial isolates against *Rhizoctonia bataticola*, a significant soil-borne pathogen. A total of 480 bacteria, 283 fungi, and 150 actinomycetes were isolated and screened using in vitro dual plate assays. Among these, isolates 5F, 131B, 223B, and 236B demonstrated the highest antagonistic activity, with inhibition rates of 88.24%, 87.5%, 81.25%, and 81.25%, respectively. The selected isolates were further assessed for abiotic stress tolerance, revealing their ability to thrive under extreme conditions. Characterization of biocontrol and plant growth-promoting activities revealed the production of siderophores, hydrogen cyanide, ammonia, chitinase, and indole-3-acetic acid, along with the solubilization of zinc and phosphorus. Compatibility tests confirmed the potential of forming effective microbial consortia, which significantly reduced the percent disease index in cluster bean. The most effective consortium, comprising *Trichoderma afroharzianum* 5F, *Pseudomonas fluorescens* 131B, *Bacillus licheniformis* 223B, and *Bacillus subtilis* 236B, achieved a 76.5% disease control. Additionally, this consortium enhanced total phenol (92.1%), flavonoids (141.6%), and antioxidant defense enzyme activities including POX (188.5%), PPOX (116.3%), PAL (71.2%), and TAL (129.9%) in cluster bean plants over the infected control, leading to substantial improvements in systemic resistance of plants. This consortium also significantly enhanced plant height, fresh weight, dry weight, number of pods per plant, and seed yield over the infected control as well as mock control. This study underscores the potential of these robust microbial consortia as a sustainable and effective strategy for managing *R. bataticola* and enhancing crop productivity under extreme environmental conditions.

## 1. Introduction

Cluster bean (*Cyamopsis tetragonoloba*), also known as guar, is a drought-resistant legume widely cultivated in arid and semi-arid regions, making it an essential crop for local agriculture [1]. Its valuable seeds are rich in guar gum, a thickening agent utilized in food, pharmaceuticals, and various industrial applications [2]. However, the cultivation of cluster bean is severely threatened by diseases caused by pathogens like *Rhizoctonia bataticola*, which can drastically reduce yields and compromise both food security and the livelihoods of farmers. Effective disease management is crucial not only for maintaining the yield and quality of cluster bean but also for ensuring the sustainability of agricultural practices in regions where this crop serves as a vital source of income.

Root rot, primarily caused by *R. bataticola*, represents a significant threat to cluster bean cultivation [3]. This pathogen is responsible for diseases such as root rot, stem rot, and seedling blight, which severely impair plant health and productivity. The initial symptoms of root rot often include yellowing of the leaves and stunted growth, which may progress to wilting, especially during dry conditions. Infected plants can develop dark, sunken lesions on the stems and roots, leading to complete plant collapse, resulting in yield losses that can reach up to 70% [4]. The substantial economic impacts of this pathogen highlight the pressing need for effective management strategies to mitigate the losses it causes.

Traditional management practices for controlling *R. bataticola* predominantly rely on the use of chemical fungicides. While these fungicides can provide temporary relief, they pose substantial environmental and health risks [5]. Continuous use of chemical fungicides can lead to the development of fungicide-resistant strains of pathogens, reduce soil biodiversity, and contaminate water sources, adversely affecting non-target organisms and human health [6,7,8]. These limitations underscore the need for sustainable and eco-friendly alternatives to chemical fungicides. Biocontrol agents, including fungi and bacteria, have emerged as promising candidates for managing *R. bataticola* due to their antagonistic activities against pathogens and their ability to promote plant growth. However, the efficacy of these biocontrol agents can be significantly influenced by environmental stress factors such as extreme temperatures, pH levels, salinity, and moisture stress [9,10,11].

There is a growing body of research indicating that biocontrol consortia, composed of multiple microbial agents, are superior to individual biocontrol agents in terms of resilience and effectiveness [12]. Consortia can offer a broader spectrum of antagonistic activities, enhanced stress tolerance, and synergistic effects, leading to improved disease control and plant health under diverse and challenging environmental conditions [13]. Microbial consortia can include a diverse range of microorganisms that vary in their environmental preferences, such as soil type, host plants, and colonization sites, as well as their activity against different pathogens [14]. Individual strains may exhibit distinct modes of action, combining multiple microorganisms within a consortium can expand the range of their antagonistic activities, providing a broader spectrum of pathogen control [15]. Moreover microorganisms within these consortia not only contribute to plant growth promotion but can also enhance the activity of other microorganisms, thereby boosting the overall efficacy of the biocontrol product [16]. A study by Minchev et al. [17] demonstrated that the combination of *Bacillus amyloliquefaciens*, *Pseudomonas chlororaphis*, and *Trichoderma harzianum* in microbial consortia provided superior protection against *Fusarium oxysporum* and *Botrytis cinerea* in tomato plants compared to individual strains. Thakkar and Saraf [18] found that soybean plants treated with a microbial consortium consisting of *Pseudomonas aeruginosa* (MBAA1), *Bacillus cereus* (MBAA2), *Bacillus amyloliquefaciens* (MBAA3), and the fungus *Trichoderma citrinoviride* (MBAAT) exhibited significantly lower disease incidence when challenged with *Macrophomina phaseolina* and *Sclerotinia sclerotiorum* compared to treatments with individual antagonists or the pathogen-infested control.

Although numerous studies have demonstrated the biocontrol potential of microbial isolates against *R. bataticola* [19,20,21], there is a paucity of research focusing on their performance under extreme abiotic stress conditions. Most existing research primarily examines the biocontrol efficacy of microbial isolates under optimal laboratory conditions, which do not accurately reflect the challenging field conditions. Furthermore, there is limited knowledge regarding the compatibility of these biocontrol agents when used in consortia, especially under such stress conditions. Therefore, evaluating the performance of biocontrol consortia under abiotic stress conditions is crucial for developing robust and sustainable plant disease management strategies.

We hypothesized that certain microbial isolates not only possess strong antagonistic activities against *R. bataticola* but also exhibit significant tolerance to extreme abiotic stress conditions. Furthermore, we hypothesized that forming microbial consortia with these robust isolates will enhance their biocontrol efficacy and plant growth-promoting (PGP) activities, leading to improved crop health and yield under stressful environmental conditions. The primary aim of this study was to identify and characterize microbial isolates that can effectively antagonize *R. bataticola* and tolerate extreme abiotic stress conditions. Additionally, we aimed to assess the compatibility and synergistic effects of selected isolates in microbial consortia and evaluate the bioefficacy of these consortia in reducing the Percent Disease Index (PDI) and enhancing the growth, biomass, and yield of cluster bean plants.

## 2. Materials and Methods

### 2.1. Isolation of Rhizospheric and Endophytic Microbes Form Plant and Soil Samples

Root, shoot, and rhizospheric soil samples were collected from cumin (*Cuminum cyminum* L.), cluster bean (*Cyamopsis tetragonoloba* [L.] Taub.), and moth bean (*Vigna aconitifolia* [Jacq.] Marechal) grown in the Jodhpur, Pali, Jaisalmer, Barmer, and Bikaner districts of India. Additionally, samples from economically significant plants in the botanical garden of ICAR-CAZRI, Jodhpur, Rajasthan, were collected. These plants included *Ocimum tenuiflorum* L., *Asparagus racemosus* Willd., *Withania somnifera* (L.) Dunal, *Euphorbia antisyphilitica* Zucc., *Cordia myxa* L., *Ephedra ciliata* Fisch. & C.A.Mey., *Opuntia cylindrica* (Lam.) DC., *Maytenus emarginata* (Willd.) Ding Hou, and *Aloe vera* (L.) Burm.f. About 20 soil samples and 20 plant samples were collected from each location for each crop. Additionally, a total of 10 soil samples and 10 plant samples were collected from the botanical garden of ICAR-CAZRI, Jodhpur. Samples from cumin were collected during the winter season (January–February), whereas samples from cluster bean and moth bean were collected during the rainy season (August–September). Samples from the botanical garden were collected during the summer season (April–May month). Soil samples were collected in zip-type poly bags. Plant samples were placed in zip-type poly bags and kept in an ice box during transport to the laboratory. Upon arrival at the laboratory, all soil and plant samples were stored at 4 °C in a refrigerator. The aim was to isolate both rhizospheric and endophytic bacteria, actinomycetes, and fungi. Fourteen different nutrient media were utilized for the isolation of microorganisms: Nutrient agar, T3 agar, Jensen agar, King’s B agar, Soil extract agar, Trypticase soy agar, R2A agar, Semi-solid nitrogen-free bromothymol malate agar, N-free Okon’s agar, Kenknight agar, Pikovskaya agar, Potato Dextrose agar, Rose Bengal agar, and Czapek Dox agar (Appendix A).

For endophyte isolation, fresh plant samples (1 g of shoot or root) were surface sterilized sequentially with 1.5% sodium hypochlorite for 5 min and 70% ethanol for 30 s [22]. The samples were then rinsed with sterile distilled water, macerated aseptically using a mortar and pestle, and serially diluted (from 10^−1^ to 10^−6^). Suitable dilutions were then spread on the respective nutrient media. For rhizospheric microorganism isolation, 1 g of soil samples was serially diluted (from 10^−1^ to 10^−8^) and individually spread on the nutrient plates. All plates were incubated at 30 °C until microbial growth was observed. Morphologically distinct colonies were further purified by sub-culturing on the respective nutrient media and stored on slants at 4 °C for subsequent analysis. Nutrient agar slants were used for storing bacterial cultures, while potato dextrose agar slants were used for storing fungal cultures.

### 2.2. Dual Plate Assay of Isolated Microbes for Antagonistic Activity Against Rhizoctonia bataticola

Isolates were tested for antagonistic activity against the phytopathogenic fungus *R. bataticola*, obtained from the Indian Type Culture Collection-Indian Agricultural Research Institute, New Delhi (ITCC Accession No-8635). The evaluation was conducted on Potato Dextrose Agar (PDA) medium using the dual culture technique in triplicate. To screen fungal isolates for antagonistic activity against *R. bataticola*, 4 mm diameter discs from actively growing cultures of both the fungal isolates and *R. bataticola* were aseptically placed near the center of a PDA plate. For screening bacterial isolates, a 4 mm diameter disc of actively growing *R. bataticola* was placed at the center of a PDA plate, and a loopful of a 24-hour-old bacterial culture was streaked equidistantly from the center on opposite sides of the disc, maintaining a 1 cm distance from the edge of the Petri dish. The plates were then incubated at 28 ± 2 °C for 5 days. Control plates contained only the *R. bataticola* mycelial disc on PDA. Antagonistic activity was assessed by comparing the mycelial diameter of *R. bataticola* on both control and test plates. The percentage of inhibition was calculated using the following formula [23]:(1)$$Percentage\;of\;Inhibition=\frac{\left(C-T\right)}{C}*100$$
where C = control mycelial diameter; T = Test mycelial diameter.

### 2.3. Assessment of Selected Microbial Isolates for Abiotic Stress Tolerance

Selected isolates were evaluated for their tolerance to a range of temperatures (30 °C, 45 °C, and 50 °C), pH levels (4, 5, 6, 7, 8, 9, and 10), salinity concentrations (1%, 5%, 10%, 15%, and 20% NaCl), and moisture stress induced by polyethylene glycol-6000 (PEG-6000) at concentrations of 5%, 10%, 15%, 20%, 25%, and 30%. To evaluate the effects of temperature, pH, salinity, and moisture stress on fungal isolates, mycelial growth (in cm) was measured on PDA plates following a 5-day incubation period, whereas bacterial growth was recorded in the form of optical density using a spectrophotometer at 630 nm after 48 h of incubation period.

For temperature tolerance, a 4 mm disc of fungal isolates was inoculated onto PDA plates and incubated at various specified temperatures for 5 days. After the incubation period, fungal growth was measured as mycelium growth (cm) [24]. For bacterial isolates, a 5% inoculum was added to nutrient broth and incubated at the respective temperatures for 48 h in a shaking incubator. Bacterial growth was recorded using a spectrophotometer at 630 nm, with an uninoculated medium serving as a blank [25].

For pH stress ability of bacteria, nutrient broth with varying pH levels was prepared. The broth was inoculated with 5% bacterial culture and incubated at 28 ± 2 °C for 48 h. Optical density (OD) was measured at 630 nm to assess bacterial growth [26]. For fungal isolates, PDA plates with different pH levels were prepared, and a 4 mm disc of fungi was placed at the center of the plates in three replicates. The plates were incubated for 5 days, and fungal growth was measured as mycelium growth (cm) [27].

The salt-tolerance ability of microbial isolates was evaluated using the methods of Zahra et al. [28]. For salt tolerance screening in bacteria, nutrient broth with varying NaCl concentrations was prepared. The broth was inoculated with 5% bacterial culture and incubated at 28 ± 2 °C for 48 h. A control containing nutrient broth without NaCl was included for comparison. OD of bacterial growth was measured at 630 nm using a spectrophotometer. For fungal isolates, PDA plates with varying NaCl concentrations were prepared, and a 4 mm disc of fungi was placed at the center of the plates in three replicates. After 5 days of incubation, fungal growth was measured as mycelium growth (cm). A control containing PDA without NaCl was included for comparison.

The moisture stress ability of microbial isolates was performed following the method described by Sandhya et al. [29]. For bacteria, nutrient broth medium with varying PEG-6000 concentrations was prepared. The broth was inoculated with 5% bacterial culture and incubated at 28 ± 2 °C for 48 h. A control containing nutrient broth without PEG-6000 was included for comparison. OD was measured at 630 nm to assess bacterial growth. For fungal isolates, PDA plates with varying PEG-6000 concentrations were prepared and inoculated with a 4 mm disc of fungi at the center of the plates in three replicates. The plates were incubated for 5 days, and fungal growth was measured as mycelium growth (cm). A control containing PDA without PEG-6000 was included for comparison.

### 2.4. Characterization of Selected Microbial Isolates for Their Biocontrol and Plant Growth-Promoting Properties

Selected microbial isolates were thoroughly characterized to evaluate their biocontrol mechanisms, focusing on their ability to produce various compounds and enzymes that enhance plant health and suppress pathogens. The characterization included the production of siderophores, hydrogen cyanide (HCN), ammonia, and hydrolytic enzymes such as chitinase, β-1,3-glucanase, cellulase, and chitosanase. The isolates were also screened for indole-3-acetic acid (IAA) production, phosphorus solubilization, and zinc solubilization. Siderophore production was assessed by spot inoculating each isolate on nutrient agar medium supplemented with chrome azurol S (CAS) dye solution. The presence of siderophores leads to a production of yellow hallow zone in the medium, indicating the ability of the isolate to chelate iron [30]. Castric’s method [31] was used for detection of HCN production by isolates. Ammonia production was detected using Dye’s method [32].

The production of chitinase, chitosanase, β-1,3-glucanase, and cellulase was determined using a well diffusion assay. Specific substrates were used for each enzyme: chitin for chitinase, glucan (laminarin) for β-1,3-glucanase, chitosan for chitosanase, and cellulose for cellulase [33,34,35,36]. For all hydrolytic enzymes, each biocontrol agent was inoculated into nutrient broth containing 1% enzyme substrate (for bacteria) or potato dextrose broth (for fungi). The cultures were incubated in a shaking incubator at 150 rpm for 72 h (bacteria) or 5 days (fungi) at 30 °C. After incubation, cultures were centrifuged at 10,000× *g* for 10 min, and the supernatant was filtered through Whatman No. 1 filter paper. The resulting cell-free broth was stored at −20 °C for further analysis. Wells of 9 mm diameter were made in 1% enzyme substrate-containing agar plates. A volume of 100 µL of each culture filtrate was added to the wells, and the plates were incubated at 37 °C for 24 h. Clear zones around the wells indicated hydrolytic activity and enzyme production [37].

Phosphorus solubilization was screened by spotting isolates on Pikovskaya’s Agar medium [38]. Zinc solubilization was tested on tris-minimal salt medium supplemented with 1% ZnO, and clear zones indicating solubilization abilities were observed after incubation at 30 °C [39]. AAA production was examined using the method of Patten and Glick [40].

### 2.5. Assessment of Compatibility Among Selected Biocontrol Agents

Isolates 5F, 131B, 223B, and 236B were cross streaked on nutrient agar plates in such a way that each isolate intersected with the others [41]. The cross-streaked plates were incubated in a biological oxygen demand (BOD) incubator at 28 ± 2 °C for 5 days. Each test was performed in triplicate to ensure reliability and reproducibility. After designated incubation period, the plates were examined for growth patterns of the *Trichoderma* and bacterial isolates. Compatibility was assessed based on the ability of the isolates to grow in close proximity without inhibiting each other’s growth. The successful integration of these isolates was determined by observing consistent growth without antagonistic effects, indicating their potential for combined use in biocontrol applications.

### 2.6. Morphological, Physiological and Biochemical Characterization of Selected Biocontrol Agents

All the selected bacterial isolates were studied for their morphological characteristics by colony characteristics, Gram staining, and KOH test. In addition, all bacterial isolates were also studied for physiological and biochemical characterization depending upon their preference for utilization of thirty-three different carbon sources viz. sucrose, lactose, melibiose, xylose, mannose, maltose, fructose, sodium gluconate, dextrose, salicin, galactose, raffinose, trehalose, arabitol, l-arabinose, inulin, erythritol, glycerol, dulcitol, inositol, sorbitol, mannitol, xylitol, adonitol, cellobiose, α-methyl-d-glucoside, rhamnose, d-arabinose, melezitose, α-methyl-d-mannoside, and sorbose and also used the carboxylate (citrate and malonate utilization). Amino acid utilization, various enzymatic activities (viz. β-galactosidase activity, esculin hydrolyses, urease, catalase phenylalanine deaminase and nitrate reductase), and H_2_S production ability of isolates were also performed. These tests were performed using HiCarboTM Kit (KB 009, HiCarbohydrate™ kit) and HiAssortedTM Biochemical Test Kit (KB002) of Himedia. All the wells of specific kits were inoculated with 50 µL of culture showing turbidity 0.5 O.D. at 630 nm and finally recording of observation on the basis of color changed after incubation of 24 to 48 h at 35–37 °C. Microscopic examination of fungal morphology, including spore shape, size, color, arrangement, hyphae, and fruiting bodies, was also determined.

### 2.7. Molecular Identification of Selected Isolates Using 16S rRNA and ITS Gene Sequencing

Four potential biocontrol agents, designated as 5F, 131B, 223B, and 236B, were selected for molecular identification through 16S rRNA and Internal Transcribed Spacer (ITS) gene sequencing techniques, targeting bacterial and fungal identification, respectively. Genomic DNA was extracted from each bacterial and fungal isolate using a DNA isolation kit (ZYMO Research Corporation, Irvine, CA, USA), adhering to the manufacturer’s instructions. For bacterial identification, the 16S rRNA gene was amplified using universal primers PA (5′-AGAGTTTGATCCTGGCTCAG-3′) and PH (5′-AAGGAGGTGATCCAGCCGCA-3′), as designed by Edwards et al. [42]. Fungal identification was achieved by amplifying the ITS region using primers ITS-1 (5′-TCCGTAGGTGAACCTGCGG-3′) and ITS-4 (5′-TCCTCCGCTTATTGATATGC-3′), following the methodology established by White et al. [43]. The purified PCR products were then Sanger sequenced by Sci-Genome Pvt. Ltd., Bangalore, India. The obtained nucleotide sequences were processed using BioEdit software (version 7.2) to trim lower-quality segments from the chromatograms. Subsequently, the trimmed sequences were compared to existing entries in the NCBI database using the BLAST tool to identify the microbes. Alignments of the sequences were performed using the CLUSTAL W program. A phylogenetic tree was constructed utilizing the bootstrap method in MEGA-11 software [44], with the Maximum Likelihood method [45] applied to infer ancestral states. The Kimura 2-parameter model [46] was used to estimate rates among sites. Finally, the identified sequences were submitted to NCBI, with accession numbers OQ152544, PQ386339, PP064158, and PQ386494 assigned to the promising isolates 5F, 131B, 223B, and 236B, respectively.

### 2.8. Pot Experiment

A pot experiment was conducted to assess the efficacy of biocontrol agents applied individually and in consortium forms, focusing on their effects on percent disease index (PDI), plant height, fresh weight, dry weight, number of pods per plant, and seed yield. Additionally, this study aimed to investigate plant–microbe interactions and the potential for induced systemic resistance (ISR). The pot trial was carried out in a glasshouse under natural environmental conditions at the ICAR-Central Arid Zone Research Institute (CAZRI), Jodhpur, Rajasthan, India. Each pot, measuring 25 cm in diameter and 25 cm in height, was filled with 10 kg of a sterilized sand–soil mixture in a 1:3 ratio. This mixture was sterilized using the tyndalization technique in an autoclave. The seeds of the susceptible cluster bean variety (Pusa Navbahar) were sourced from the Division of Plant Improvement and Pest Management at ICAR-CAZRI, Jodhpur. Prior to planting, the seeds underwent surface sterilization with 70% ethanol for 1 min, followed by treatment with 1.5% sodium hypochlorite solution for 5 min [17].

Bacterial isolates (131B, 223B, and 236B) were cultured in nutrient broth and incubated at 30 °C, 160 rpm for 1 day. *Trichoderma* isolate 5F was cultured in potato dextrose broth and incubated at 30 °C, 160 rpm for 7 days. Upon the completion of the incubation period, concentrations reached 10⁸ CFU mL^−1^ for bacteria and 10⁴ spores mL^−1^ for *Trichoderma*. To apply the microbial consortia, suspensions of each microorganism were prepared at specific concentrations: *Trichoderma* was set at 10^4^ spores mL^−1^, while bacterial suspensions were prepared at 10^8^ CFU mL^−1^. These individual suspensions were then combined in equal proportions to create the consortia, following the treatments as outlined in the experimental design.

Seeds were treated with biocontrol agent suspensions, using charcoal as a carrier medium. Seeds were soaked in the suspension for 15 min and air-dried for 2 h. For comparative analysis, control seeds were treated with sterile water and charcoal. After treatment, the seeds were sown in pots and thinned to six plants per pot following germination. Recommended doses of fertilizers (RDFs) were applied according to specific ratios of nitrogen and phosphorus, with 20 kg N ha^−1^ and 40 kg P_2_O_5_ ha^−1^.

The experiment included the following eight treatments, conducted in a Completely Randomized Design with six replications: T_1_: No pathogen + No biocontrol agent (mock control); T_2_: Only *R. bataticola* (infected control); T_3_: *Trichoderma afroharzianum* 5F + challenged with *R. bataticola;* T_4_: *Pseudomonas fluorescens* 131B + challenged with *R. bataticola;* T_5_: *Bacillus licheniformis* 223B + challenged with *R. bataticola;* T_6_: *Bacillus subtilis* 236B + challenged with *R. bataticola;* T_7_: 131B + 223B + 236B + challenged with *R. bataticola;* T_8_: 5F + 131B + 223B + 236B + challenged with *R. bataticola.* Three replications were used to collect the data on PDI, number of pods per plant, and seed yield. The remaining three replications were used for analyzing plant height, fresh weight, dry weight, and biochemical characteristics of cluster bean.

Biocontrol agents were applied at a rate of 10 g kg^−1^ of soil using a talc-based formulation at intervals of 20, 30, and 40 days after sowing (DAS). The pathogen *R. bataticola* was introduced through drenching treatments T2 to T8 on the 50th DAS. The inoculum for *R. bataticola* was prepared by culturing the pathogen on autoclaved sorghum and applied at a rate of 10 g/kg of soil, which contained 10⁴ spores g^−1^ [47].

#### 2.8.1. Percent Disease Incidence

To assess the effectiveness of the biocontrol agents, Percent Disease Incidence (PDI) and Percent Disease Control (PDC) were calculated. Weekly observations were made to record the symptoms and the number of surviving plants for the pathogen-infected individuals across the various treatments maintained in the glasshouse. The following formulas were used to determine PDI and PDC [48]:(2)$$PDI=\frac{Number\;of\;infected\;plants}{Total\;number\;of\;plants}\times 100$$
(3)$$PDC=\frac{PDI\;of\;control-PDI\;of\;treatment}{PDI\;of\;control}\times 100\;$$

#### 2.8.2. Estimation of Total Phenol and Flavonoid

The total phenol in the methanolic extracts of cluster bean leaves was measured using the Folin–Ciocalteu reagent (FCR) method, with modifications based on Dewanto et al. [49]. A 200 µL of extract from 500 mg of cluster bean leaves was combined with 2.5 mL of 0.2 M FCR and allowed to stand for 5 min at room temperature. Subsequently, 2 mL of 7.5% sodium carbonate was added, and the mixture was thoroughly mixed before being incubated for 90 min at 25 °C. The absorbance was measured at 760 nm using a UV-VIS spectrophotometer (Systronics, Ahmedabad, India). Total phenolic content was quantified using calibration curve prepared with catechol and expressed as mg of gallic acid equivalents per gram of fresh weight (mg GAE g^−1^ F.W.).

In addition to phenolic compounds, the total flavonoid content was determined using the aluminum chloride method as outlined by Zou et al. [50]. A 1.5 mL of methanolic extract from 500 mg of cluster bean leaves was mixed with 75 µL of a 2% (*w*/*v*) aluminum chloride solution and 0.5 mL of sodium acetate solution. Thereafter, final volume was adjusted to 2.5 mL using distilled water. After a 30 min incubation period at room temperature in the dark, the absorbance was recorded at 415 nm against a methanol blank. Quercetin was used as a standard for quantification (0–50 µg mL^−1^), and the results were expressed as mg quercetin equivalents per gram of fresh weight (mg QE g^−1^ F.W.).

#### 2.8.3. Quantitative Determination of Antioxidant-Defense Enzymes

For the analysis of peroxidase (POX) and polyphenol oxidase activity (PPOX), one gram of fresh leaves was ground in 10 mL of ice-cold 0.05 M potassium phosphate buffer (pH 7.0), supplemented with 0.5 mM EDTA. The homogenate was then centrifuged at 10,000× *g* for 15 min at 4 °C, and the resulting supernatant was used as the enzyme extract. POX activity was determined using the method described by Shannon et al. [51]. The specific activity of the enzyme was expressed as U min^−1^ g^−1^ F.W. The assessment of PPOX activity was carried out according to the method described by Jockusch [52] and expressed as U min^−1^ g^−1^ F.W.

To assess Phenylalanine/Tyrosine ammonia lyase (PAL/TAL) activity, 500 mg of fresh leaves were macerated in 5 mL of an ice-cold extraction buffer composed of 0.2 M sodium borate and 0.08% β-mercaptoethanol. The homogenate was then centrifuged at 10,000× *g* for 15 min at 4 °C, with the supernatant serving as the enzyme extract for PAL/TAL analysis. PAL activity was measured using the protocol by Mahatma et al. [53] and expressed as U h^−1^ g^−1^ F.W. TAL activity was assessed using the method of Mahatma et al. [54] and expressed as units per hour per gram of fresh weight (U h^−1^ g^−1^ F.W.).

#### 2.8.4. Measurement of Plant Height, Biomass, Yield and Yield Attributes

To measure plant height, fresh weight, and dry weight, three plants were uprooted from each replication after 70 DAS. For dry weight determination, samples were placed in paper envelopes and initially dried in an oven at 105 °C for two hours, followed by continuous drying at 80 °C until a consistent dry weight was achieved. At maturity, the number of pods plant^−1^ was counted. Furthermore, the seed yield of cluster beans from each plant, under various treatments, was carefully weighed and recorded in gram pot^−1^.

### 2.9. Statistical Analysis

The data obtained from the experiments were utilized to analyze the mean values from three replications of each treatment. Statistical analysis was conducted using Minitab 17 statistical software. To evaluate the grouping information between the mean values from each experiment, the Fisher’s Least Significant Difference (LSD) method was applied, using a 95% confidence level (*p* ≤ 0.05) to determine significant differences between the treatments.

## 3. Results

### 3.1. Dual Plate Assay of Isolated Microbes for Antagonistic Activity Against Rhizoctonia bataticola

A total of 480 bacteria, 283 fungi, and 150 actinomycetes were isolated and screened against *R. bataticola* in dual plate assay. The in vitro antagonistic assay of fungal isolates against *R. bataticola* demonstrated varying degrees of inhibition across different isolates. Among the fungal isolates, only 22 isolates showed in vitro antagonistic activity against *R. bataticola.* Isolates 1F and 5F both exhibited the highest inhibition, with an 88.24% reduction in the mycelial growth of *R. bataticola* (Figure 1). Similarly, isolate 37F demonstrated substantial inhibition, with a 76.47% reduction in mycelial growth. Isolate 42F also displayed strong antagonistic activity with an inhibition percentage of 82.35%. Other notable isolates include 7F, which showed a 67.06% inhibition (Appendix A).

The in vitro antagonistic assay of various bacterial and actinomycete isolates against *R. bataticola* revealed a broad spectrum of inhibition percentages, reflecting the diverse antagonistic potentials among the isolates (Appendix A). Notably, isolate 131B exhibited the highest inhibition rate at 87.5%, followed by isolates 223B and 236B, each showing 81.25% inhibition (Figure 1). Additionally, isolates 9B, 32B, 78B, 143B, 158B, and 193B all demonstrated 75% inhibition, while isolates 48B and 391B also showed strong antagonistic activity with 72.5% inhibition. Conversely, some isolates displayed minimal antagonistic activity, such as 18B with only 2.5% inhibition, and isolates 4B, 11B, 27B, 70B, and 71B, which exhibited inhibition percentages around 6.25%. Several isolates showed moderate levels of inhibition, ranging from 50% to 70%, including isolates 10B, 16B, 72B, and 100B. Based on their antagonistic activity, isolates 1F, 5F, 37F, 42F, 131B, 223B, 236B, 242B, 325B, and 390B were selected for further study.

### 3.2. Assessment of Selected Fungal Isolates for Abiotic Stress Tolerance

The mycelial growth of four fungal isolates (1F, 5F, 37F, and 42F) was assessed under varying conditions of temperature, pH, salinity, and polyethylene glycol (PEG-6000). The results revealed that all fungal isolates exhibited optimal growth at moderate temperatures (30 °C and 45 °C) and at neutral to slightly acidic pH levels (pH 5 to 7). Higher temperatures (50 °C) and extreme pH levels (pH 9 and 10) significantly inhibited mycelial growth. Notably, isolate 5F showed maximum growth at 50 °C among all the isolates (Figure 2). Under salinity and moisture stress conditions, all fungal isolates demonstrated optimal mycelial growth at low salinity (1% to 10% NaCl) and low moisture stress (5% to 10% PEG). Growth significantly decreased at higher salinity levels (10% NaCl and above) and higher moisture stress levels (15% PEG and above), with no growth observed at the highest salinity level (20% NaCl). Minimum mycelial growth was recorded at 15% NaCl and 30% PEG (Figure 3).

### 3.3. Assessment of Selected Bacterial Isolates for Abiotic Stress Tolerance

The absorbance of six bacterial isolates (131B, 223B, 236B, 242B, 325B, and 390B) was assessed under varying conditions of temperature, pH, salinity, and polyethylene glycol (PEG-6000). Results revealed that all bacterial isolates showed optimal growth within the temperature range of 30 °C to 45 °C. However, at 50 °C, the growth of all isolates declined, with isolate 223B recording the highest absorbance at 0.85, followed closely by 236B at 0.83 (Figure 4). In contrast, isolates 131B and 242B exhibited the lowest growth at 0.60 and 0.60, respectively. Regarding pH levels, bacterial isolates demonstrated optimal growth between pH 5 and 7, with robust growth observed at pH 7. Growth progressively decreased above pH 7, and at extreme pH levels (pH 4 and pH 10), significant reductions were noted, with absorbance values dropping to as low as 0.13 ± 0.04 for most isolates at pH 10. Isolates 223B and 236B showed relatively more growth at these extreme pH levels compared to the other isolates (Figure 4).

Under salinity stress, bacterial isolates demonstrated considerable tolerance up to 10% NaCl, showing high absorbance at 1% NaCl. Although absorbance values decreased with increasing salinity, isolates 223B and 236B maintained robust growth at 5% and 10% NaCl. At 15% NaCl, absorbance values dropped significantly, with isolates 223B and 236B showing maximum absorbance of 0.6 compared to the other isolates. At the highest salinity level of 20% NaCl, only isolates 223B and 236B showed any growth, with absorbance values of 0.2 (Figure 5). In response to PEG-6000-induced moisture stress, bacterial isolates exhibited a gradual decrease in absorbance as PEG concentration increased. At 5% PEG, all isolates displayed high absorbance. The decline in growth became more pronounced at 30% PEG. Interestingly, at 30% PEG, isolates 223B and 236B showed the highest absorbance values of 0.47 and 0.57, respectively. Overall, isolates 223B and 236B demonstrated higher absorbance values across a wide range of temperatures, pH levels, salinity, and moisture stress, followed by isolate 131B (Figure 5).

### 3.4. Characterization of Selected Microbial Isolates for Their Biocontrol and Plant Growth-Promoting Properties

The selected fungal and bacterial biocontrol agents exhibited a remarkable range of beneficial characteristics, including the production of siderophores, hydrogen cyanide (HCN), ammonia, chitinase, beta-1,3-glucanase, chitosanase, and indole-3-acetic acid (IAA). Additionally, they demonstrated the ability to solubilize zinc (Zn) and phosphorus (P). Notably, all bacterial isolates tested negative for cellulase activity. Among the biocontrol agents, isolates 5F, 131B, 223B, and 236B displayed the most promising biocontrol and plant growth-promoting (PGP) activities, leading to their selection for further investigation (Table 1).

### 3.5. Assessment of Compatibility Among Selected Biocontrol Agents

Four promising biocontrol agents—designated as 5F, 131B, 223B, and 236B—were chosen due to their notable effectiveness in both in vitro antagonistic assays and plant growth-promoting (PGP) activities. The compatibility tests conducted using cross-streaking methods demonstrated that these microbial isolates can coexist harmoniously, with no evidence of growth inhibition among them (Figure 6). The results from the cross-streaking assays confirmed that all selected microbes were capable of growing together without any detrimental effects on each other’s growth.

### 3.6. Morphological, Physiological and Biochemical Characterization of Selected Biocontrol Agents

The bacterial isolates 236B, 223B, and 131B exhibited distinct morphological characteristics. Isolate 236B was medium-sized with a rod shape, irregular margins, and an opaque appearance. Its elevation was flat, and it had a dry, rough texture with a creamy pigmentation. Isolate 223B, also rod-shaped, was large with lobate margins, opaque opacity, and a raised elevation, exhibiting a dry, rough texture and white pigmentation. In contrast, isolate 131B was small and rod-shaped with undulate margins, translucent opacity, and a convex elevation. This isolate had a mucoid texture and displayed a greenish-yellow pigmentation (Appendix A).

The physiological and biochemical characterization of the bacterial isolates 236B, 223B, and 131B revealed distinct metabolic profiles. Isolate 236B could utilize xylose, dextrose, galactose, melibiose, L-arabinose, mannose, and sorbitol, and it was positive for citrate and malonate utilization, oxidase, casein hydrolysis, and catalase tests. Isolate 223B showed utilization of dextrose, inulin, sodium gluconate, glycerol, salicin, inositol, sorbitol, erythritol, α-methyl-d-glucoside, rhamnose, cellobiose, melezitose, α-methyl-d-mannoside, xylitol, and sorbose, and it was positive for citrate and malonate utilization, ONPG, esculin hydrolysis, oxidase, casein hydrolysis, and catalase tests. Isolate 131B could utilize lactose, xylose, maltose, fructose, dextrose, galactose, raffinose, trehalose, melibiose, sucrose, L-arabinose, mannose, adonitol, xylitol, and rhamnose, and it was positive for citrate utilization, ONPG, urease, nitrate reductase, oxidase, and catalase tests, as well as the KOH test. Notably, 236B and 223B were Gram-positive, while 131B was Gram-negative (Appendix A).

The fungal isolate 5F, identified as *Trichoderma afroharzianum*, exhibited distinct microscopic and macroscopic characteristics. The spores were globose to subglobose in shape and exhibited a greenish color. They were arranged in clusters at the tips of the conidiophores. The hyphae were septate, hyaline, and smooth in texture. The fruiting bodies featured branched conidiophores with flask-shaped phialides (Appendix A).

### 3.7. Molecular Identification of Selected Isolates Using 16S rRNA and ITS Gene Sequencing

Through 16S rRNA and ITS gene sequencing, followed by phylogenetic analysis using the NCBI database, the isolates 5F, 131B, 223B, and 236B showed 100%, 100%, 99.53%, and 100% similarity with *Trichoderma afroharzianum* strain TBR-4 (Accession no.MN944478), *Pseudomonas fluorescens* CFLB-16 (Accession no. ON764429), *Bacillus licheniformis* strain B21 (Accession no. MK583660), and *Bacillus subtilis* strain 181203-033_G01_4DD (Accession no. MT448935), respectively (Figure 7). The gene sequences of 5F, 131B, 223B, and 236B have been submitted to NCBI with accession no. OQ152544, PQ386339, PP064158, and PQ386494, respectively.

### 3.8. Bioefficacy Evaluation of Selected Biocontrol Agents Against Rhizoctonia bataticola

The evaluation of the bioefficacy of selected biocontrol agents against *R. bataticola* in cluster bean pot experiments revealed notable findings (Table 2). In the mock control with no pathogen and no biocontrol agent, the Percent Disease Index (PDI) was minimal at 11.0. In contrast, the infected control group exhibited a significantly high PDI of 94.44. Among the individual biocontrol treatments, application of *T. afroharzianum* 5F showed the more significant results with a PDI of 55.55%, achieving 41.2% disease control. The consortium treatments demonstrated enhanced bioefficacy. Application of consortia containing *P. fluorescens* 131B, *B. licheniformis* 223B, and *Bacillus subtilis* 236B significantly reduced the PDI to 38.88 ± 8.7, achieving 58.8% disease control. The most effective treatment was the consortium of *T. afroharzianum* 5F, *P. fluorescens* 131B, *B. licheniformis* 223B, and *B. subtilis* 236B, which significantly lowered the PDI to 22.2, resulting in 76.5% disease control.

### 3.9. Impact of Individual and Consortium Applications of Biocontrol Agents on Biochemical Characteristics and Antioxidant Defense Enzymes in Cluster Bean Against Rhizoctonia bataticola

In a pot experiment assessing the impact of biocontrol agents on the biochemical characteristics and antioxidant-defense enzymes of cluster bean plants challenged with *R. bataticola*, notable differences were observed across treatments (Table 3). The mock control (T1), which had no pathogen or biocontrol agents, exhibited the lowest levels of total phenols, flavonoids, and antioxidant-defense enzymes. In contrast, the infected control showed increased levels in these parameters compared to the mock control. Among the individual biocontrol agents, *T. afroharzianum* 5F (T3) demonstrated the highest levels of total phenols, flavonoids, and antioxidant enzymes. *T. afroharzianum* 5F led to a 39.5% increase in total phenols compared to the infected control. Similarly, this treatment resulted in a 100% increase in POX activity, a 52.3% increase in PPOX, a 36.5% increase in PAL, and a 62.2% activity increase in TAL. *P. fluorescens* 131B, *B. licheniformis* 223B, and *B. subtilis 236B* also significantly enhanced these parameters compared to the infected control, though to a lesser extent.

The consortium treatments showed the most substantial improvements. Application of three bacterial consortia including *P. fluorescens* 131B, *B. licheniformis* 223B, and *B. subtilis* 236B showed an impressive 89.5% increase in total phenols, 91.7% in flavonoids, 134.6% in POX, 102% in PPOX, 51.9% increase in PAL, and a 84% increase in TAL activity. The four microbe consortium treatment, which included *T. afroharzianum* 5F, *P. fluorescens* 131B, *B. licheniformis* 223B, and *B. subtilis* 236B, resulted in the highest enhancements, with a 92.1% increase in total phenols, 141.6% in flavonoids, a 188.5% increase in POX activity, a 116.3% increase in PPOX, a 71.2% increase in PAL, and a 129.9% increase in TAL.

### 3.10. Impact of Biocontrol Agents on Growth, Biomass, Yield, and Yield Attributes of Cluster Bean Plants Challenged with Rhizoctonia bataticola

In a pot experiment evaluating the impact of biocontrol agents on cluster bean growth and yield under *R. bataticola* infection, significant differences were observed with various treatments (Table 4). The infected control exhibited severely reduced growth and yield. Among the biocontrol agents tested individually, *T. afroharzianum* 5F was effective in improving plant height, fresh weight, dry weight, number of pods, and yield compared to the infected control, though not as markedly as the consortium treatments. Similarly, *P. fluorescens* 131B (T4), *B. licheniformis* 223B (T5), and *B. subtilis* 236B (T6) also enhanced plant attributes but did not achieve the same level of improvement as the consortium treatments. However, there was no significant difference among the individual biocontrol agents in relation to plant height, fresh weight, dry weight, whereas *T. afroharzianum* 5F significantly enhanced the yield compared to other individually used biocontrol agents.

The most significant results were recorded with the consortia treatments. The combination of three bacteria, namely-*P. fluorescens* 131B, *B. licheniformis* 223B, and *B. subtilis* 236B, significantly improved all growth parameters compared to the infected control, with increases in plant height, fresh weight, dry weight, number of pods, and yield. The most pronounced improvement was seen with four microbe consortium including *T. afroharzianum* 5F, *P. fluorescens* 131B, *B. licheniformis* 223B, and *B. subtilis* 236B. This treatment resulted in the highest values for all parameters, demonstrating a substantial enhancement in plant growth and yield. Specifically, plant height increased by 50.5%, fresh weight by 116.6%, dry weight by 116.7%, number of pods by 7-fold, and yield by 28-fold compared to the infected control. In addition, this consortium also significantly enhanced plant height, fresh weight, dry weight, number of pods per plant, and yield per pot by 21.25%, 20.32%, 18.18%, 10.5%, and 12.81%, respectively, over the mock control.

## 4. Discussion

The management of *R. bataticola*, a pathogen responsible for significant crop losses, is critical for sustainable agriculture. Traditional chemical methods pose environmental and health risks, prompting the need for eco-friendly alternatives such as biocontrol agents. This study investigated the potential of fungal and bacterial isolates to antagonize *R. bataticola*, their abiotic stress tolerance, PGP activities, and their compatibility when used in consortia. The in vitro antagonistic assay revealed that specific isolates exhibited significant antagonistic activity against *R. bataticola*. Among the fungal and bacterial isolates, 1F, 5F, 131B, 223B, and 236B demonstrated the highest inhibition of pathogen. This suggests a strong potential for biocontrol, likely due to the production of antifungal metabolites (siderophore, HCN, ammonia, chitinase, beta-1, 3-glucanase, cellulase, chitosanase) or competitive exclusion mechanisms.

These findings are aligned with previous studies where microbial isolates with high antagonistic activity were effective in pathogen suppression through mechanisms such as competition for nutrients and space, production of antifungal compounds, and induction of plant resistance [55]. Present findings also support those of Kumari et al. [56], who found that the mycelial growth of test pathogens was inhibited by *Trichoderma* spp. Due to the release of various diffusible volatiles and non-volatile compounds in the medium, such as harzianic acid, heptelidic acid, tricholin, and glisoprenins. Recently, endophytic antagonistic *B. subtilis* has been shown to inhibit the phytopathogen *R. bataticola*, leading to morphological changes in fungal hyphae, including bursting, swelling, and lysis [57].

In the present study, the selected fungal isolates and bacterial isolates were able to tolerate a wide range of temperature pH, salinity, and moisture stress. The decline in growth at higher temperatures and extreme pH levels is consistent with the physiological limits of most microbes, where enzyme denaturation and metabolic disruptions occur [58]. Isolate 5F’s ability to grow at 50 °C indicates its potential for application in high-temperature environments. For bacterial isolates, 223B and 236B showed the highest tolerance to extreme salinity and moisture stress, maintaining growth at 15% NaCl and 30% PEG-6000. This tolerance is crucial for biocontrol efficacy in arid and saline soils where *R. bataticola* thrives. The observed stress tolerance can be attributed to the production of osmoprotectants, exopolysaccharides, and stress-related proteins [59].

The selected isolates exhibited several PGP activities, including siderophore production, phosphate solubilization, and synthesis of IAA. These traits contribute to enhanced nutrient availability and growth regulation in plants [60]. For example, siderophore production facilitates iron uptake in plants, crucial for chlorophyll synthesis and enzymatic functions [61]. The production of chitinase, beta-1,3-glucanase, and chitosanase indicates the potential of these isolates to degrade fungal cell walls, thus providing direct antagonism against *R. bataticola* [62,63]. Production of auxins is an important trait when screening PGP microorganisms [64]. In the present study, all selected antagonistic bacteria and fungi showed the production of auxins. Auxins are vital for root initiation and development, enhancing root surface area for nutrient and water uptake, and help plants cope with abiotic stresses by modulating stress-responsive genes, thereby improving growth and yield under adverse conditions [65].

The compatibility tests showed that the selected biocontrol agents (5F, 131B, 223B, 236B) could coexist without inhibiting each other’s growth. This suggests a potential for forming a microbial consortium that can function synergistically, enhancing overall biocontrol efficacy. Coexistence without antagonism among biocontrol agents is essential for the stability and effectiveness of microbial consortia in field applications [13]. Earlier studies have been performed cross-streaking compatibility between microbial isolates for development of microbial consortia for waste decomposition, plant growth promotion, and biocontrol [66,67].

The identification of the promising isolates through 16S rRNA and ITS gene sequencing confirmed their taxonomic positions: *T. afroharzianum* 5F, *P. fluorescens* 131B, *B. licheniformis* 223B, and *B. subtilis* 236B. These species are well-documented for their biocontrol and PGP activities. *Trichoderma* spp. are known for their mycoparasitism, antibiosis, and induction of systemic resistance in plants [68,69]. *P. fluorescens* is recognized for its ability to produce antifungal metabolites and enhance plant growth through various mechanisms [70,71]. *Bacillus* species are noted for their resilience to environmental stress and production of a wide range of antifungal compounds [72,73].

The present results demonstrated significant disease control by the individual and consortium treatments. The consortium of *T. afroharzianum* 5F, *P. fluorescens* 131B, *B. licheniformis* 223B, and *B. subtilis* 236B showed the highest reduction in the PDI, achieving 76.5% disease control. This superior efficacy of the consortium can be attributed to the complementary mechanisms of action, such as enhanced production of antifungal metabolites, increased nutrient availability, and induction of systemic resistance in plants. The synergistic effect of microbial consortia has been reported to provide more robust and consistent disease control compared to individual agents [12].

Our results are corroborated with Dhawan et al. [74] who reported that application of *T. viride* 10 g/kg soil reduced the dry root rot infestation in cluster bean by 34.8% over the uninoculated control. Singh et al. [75] explored the efficacy of *T. harzianum. T. Viride, T. hematum, glicladimvire, P. fleurescens*, and *B. subtilis* for managing the dry root rot of chickpea and significantly controlled dry root rot disease. Senthilkumar et al. [20] also reported that application of *Paenibacillus* sp. HKA-15 in soybean significantly reduced the percent disease incidence of *R. bataticola* which showed only 20% PDI.

The biocontrol agents, particularly in consortium mode, significantly enhanced the biochemical characteristics and antioxidant defense enzymes of cluster bean plants. The increase in total phenols, flavonoids, and antioxidant enzymes (POX, PPO, PAL, TAL) indicates the induction of systemic resistance and mitigation of oxidative stress in plants. The enhanced levels of these compounds are crucial for strengthening plant defense mechanisms against pathogen attack [76]. The consortium treatments showed the highest increases, suggesting a more robust activation of plant defense pathways due to the combined effects of the biocontrol agents and contributing to increased resistance against pathogen attack and improved overall plant health. Hydrogen cyanide (HCN), ammonia, and siderophores produced by biocontrol agents can act as elicitors of induced systemic resistance (ISR) in plants, resulting in enhanced secretion of phenols, flavonoids, and antioxidant defense enzymes [77,78,79]. Therefore, beneficial microbes play a significant role in plant-microbe interactions and defense signaling.

The findings of the present study align with those of Kumar et al. [80] who demonstrated that the application of a consortium of *Bacillus subtilis* and *Trichoderma harzianum* significantly enhanced tomato growth and defensive responses against the pathogens *Alternaria solani* and *Phytophthora infestans* by increasing the activity of antioxidant-defense enzymes like polyphenol oxidase, peroxidase, and superoxide dismutase. Recently Singh et al. [81] reported that application of consortia comprising *Trichoderma atrobruneum* 15F, *Pseudomonas* sp. 2B, *Bacillus amyloliquefaciens* 9B, and *Bacillus velezensis* 32B significantly increased the level of total phenols, flavonoids, and the activities of antioxidant-defense enzymes-POX, PPOX, PAL, and TAL in cumin plants challenged with *Fusarium oxysporum* f. sp. *Cumin.*

In the present study, the impact of biocontrol agents on plant growth and yield was most pronounced in the consortium treatments, which significantly improved plant height, fresh weight, dry weight, number of pods, and yield compared to the infected control as well as mock control. These enhancements can be attributed to the combined PGP activities of the biocontrol agents, such as nutrient solubilization, hormone production, and disease suppression, which collectively contribute to improved plant health and productivity [82]. Gautam et al. [83] investigated that the colonization of roots by biocontrol agents not only suppressed disease but also enhanced plant growth by increasing nutrient uptake and releasing plant hormones. Additionally, Srivastava et al. [84] reported that the interaction of *Trichoderma* spp. with plants promotes growth, enhances nutrient availability, increases crop yield, and improves disease resistance.

The results of the present study are aligned with earlier studies where an increase in the shoot length, with the biocontrol agents has been reported in maize [85,86]. Choudhary and Ashraf [87] reported that plant height, fresh weight, number of pods per plant, test weight, and seed yield were significantly enhanced with application of *T. harzianum* in mung bean against dry root rot. The results on the plant growth in the present study are also corroborated with Vetrivelkalai et al. [88] who reported that inoculation of different strains of *Pseudomonas* sp., *Bacillus* sp., and *Methylobacterium* sp. varied significantly in enhancing the plant growth of bhendi seedlings.

## 5. Conclusions

The present study successfully identified and characterized several promising microbial biocontrol agents, including the fungal isolate *T. afroharzianum* 5F and bacterial isolates *P. fluorescens* 131B, *B. licheniformis* 223B, and *B. subtilis* 236B. These isolates exhibited significant antagonistic activity against *R. bataticola*, with high inhibition rates highlighting their potential for effective pathogen management. Notably, both fungal and bacterial isolates demonstrated considerable tolerance to various abiotic stress conditions, such as temperature, pH, salinity, and moisture stress, indicating their adaptability to diverse environmental scenarios. The application of these selected biocontrol agents significantly enhanced the biochemical characteristics and antioxidant defense enzymes in cluster bean plants. The consortium treatments comprising *T. afroharzianum* 5F, bacterial isolates *P. fluorescens* 131B, *B. licheniformis* 223B, and *B. subtilis* 236B yielded the most substantial increases in total phenols, flavonoids, and key antioxidant enzymes, suggesting their critical role in bolstering the plant’s defense mechanisms against pathogens. Furthermore, this consortium treatment outperformed individual applications in enhancing plant growth and yield parameters. These findings support the development of effective biocontrol strategies utilizing microbial consortia, which can enhance plant resilience against pathogens while promoting growth. This approach aligns with sustainable agricultural practices, reducing reliance on chemical pesticides and fostering healthier ecosystems. Future research should focus on conducting field trials to validate the efficacy of these biocontrol agents in real agricultural settings and further explore their mechanisms of action and potential synergistic interactions within various crop systems.

## Figures and Tables

**Figure 1 microorganisms-12-02331-f001:**
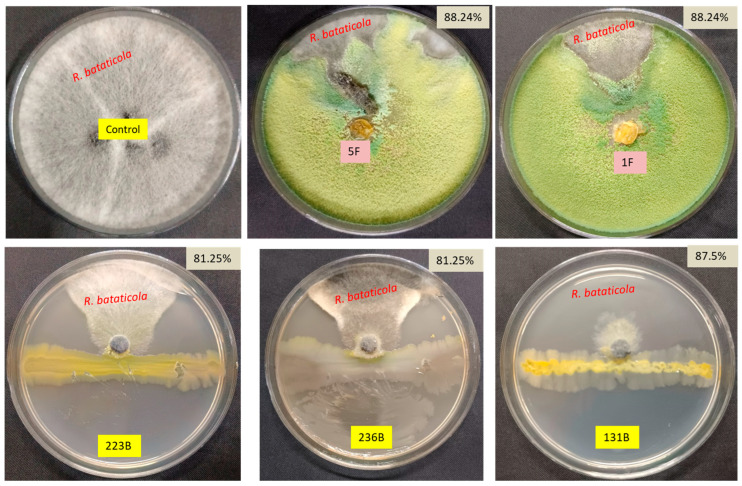
In vitro antagonistic assay of fungal and bacterial isolates against *Rhizoctonia bataticola*.

**Figure 2 microorganisms-12-02331-f002:**
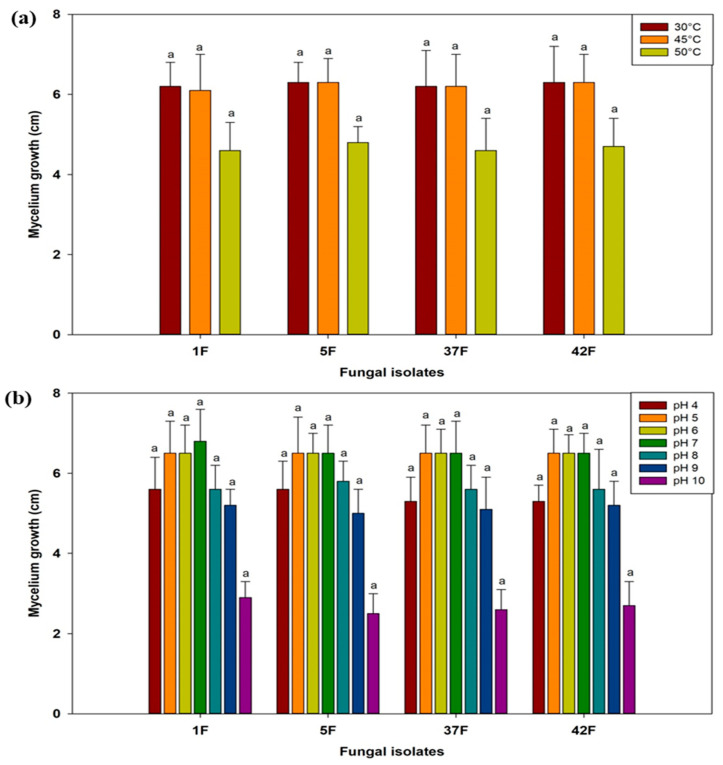
Analysis of temperature (**a**) and pH (**b**) stress tolerance in selected fungal isolates. Data are the average of three replicates. Error bars show the SD. Different letters point out significant differences among the fungal isolates or bacterial isolates for individual temperature and pH.

**Figure 3 microorganisms-12-02331-f003:**
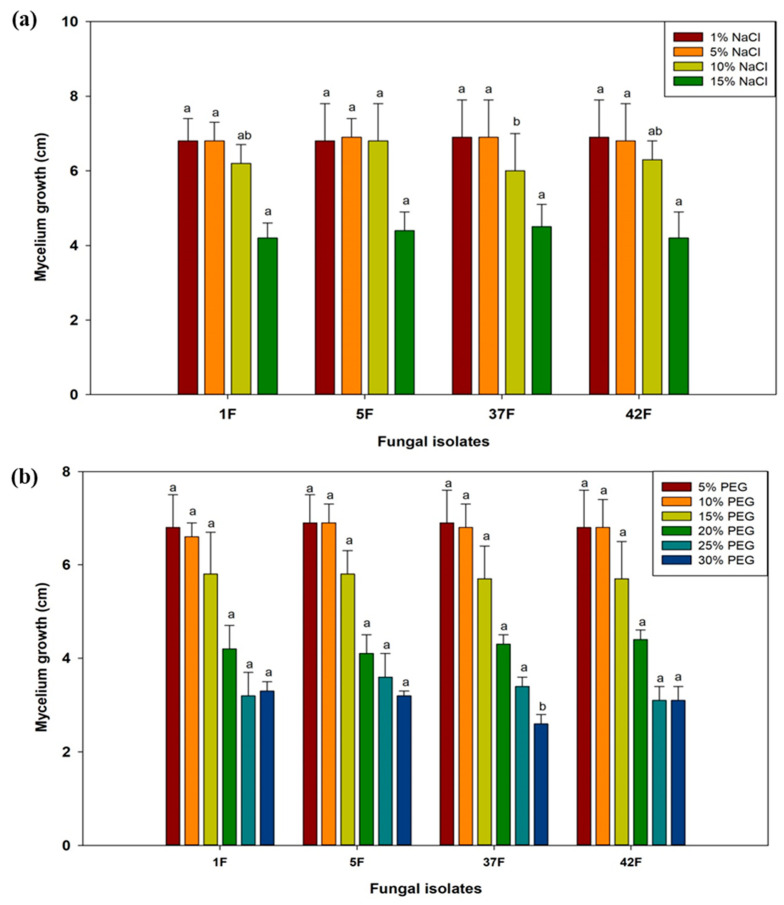
Analysis of salinity (**a**) and moisture (**b**) stress tolerance in selected fungal isolates. Data are the average of three replicates. Error bars show the SD. Different letters point out significant differences among the fungal isolates or bacterial isolates for individual NaCl and PEG-6000 concentration.

**Figure 4 microorganisms-12-02331-f004:**
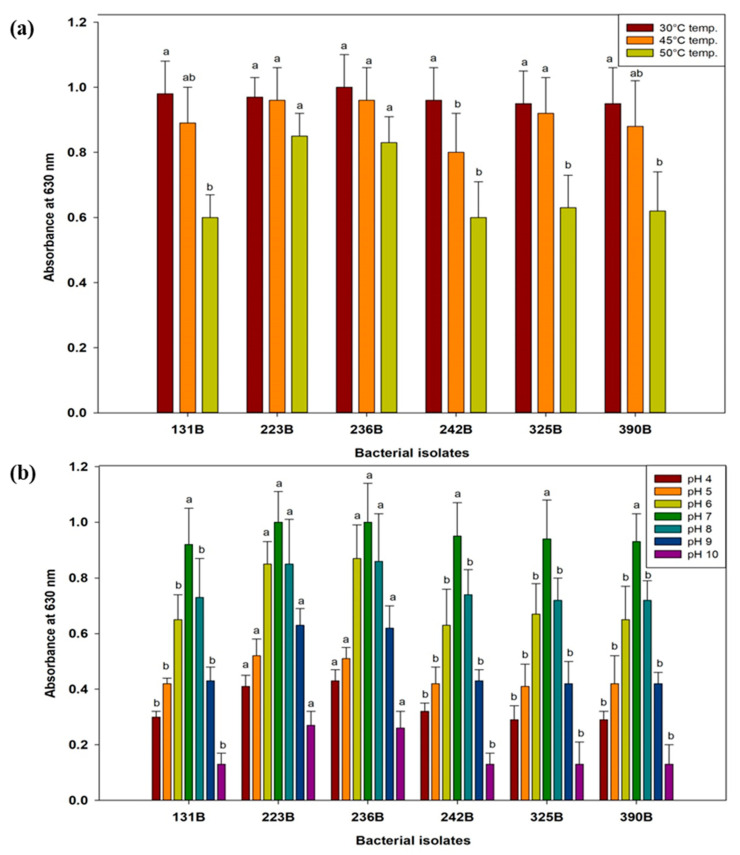
Analysis of temperature (**a**) and pH (**b**) stress tolerance in selected bacterial isolates. Data are the average of three replicates. Error bars show the SD. Different letters point out significant differences among the fungal isolates or bacterial isolates for individual temperature and pH.

**Figure 5 microorganisms-12-02331-f005:**
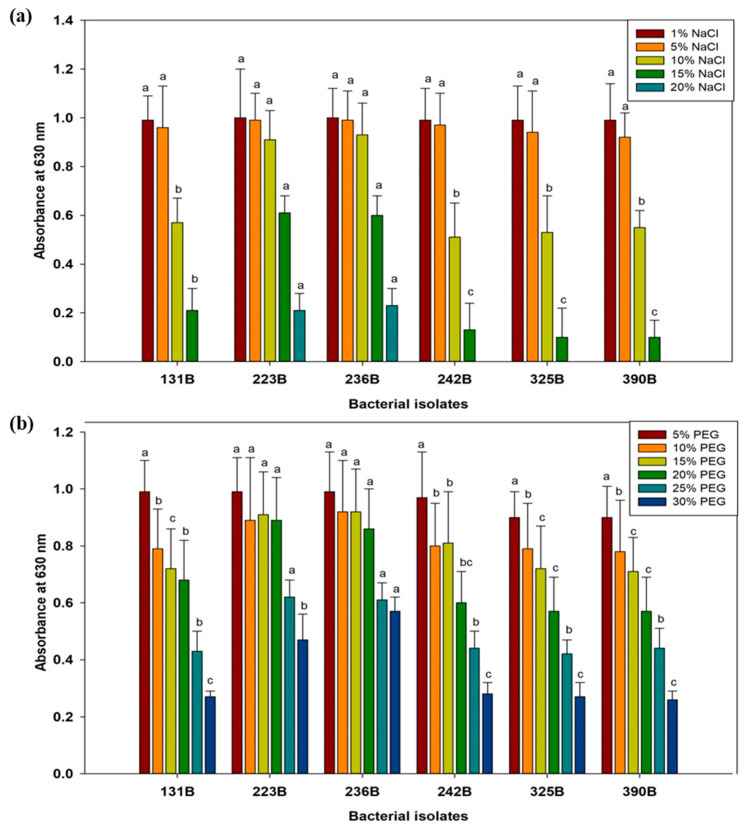
Analysis of salinity (**a**) and moisture stress (**b**) tolerance in selected bacterial isolates. Data are the average of three replicates. Error bars show the SD. Different letters point out significant differences among the fungal isolates or bacterial isolates for individual NaCl and PEG-6000 concentration.

**Figure 6 microorganisms-12-02331-f006:**
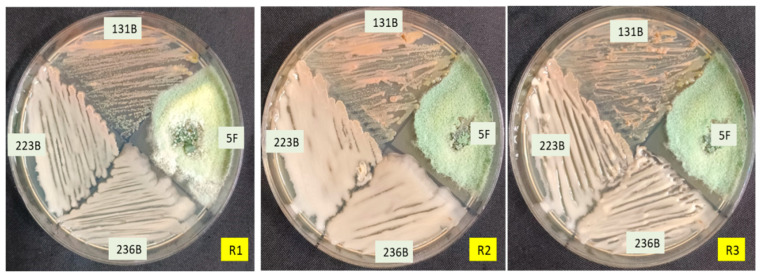
In vitro compatibility assessment of selected biocontrol agents.

**Figure 7 microorganisms-12-02331-f007:**
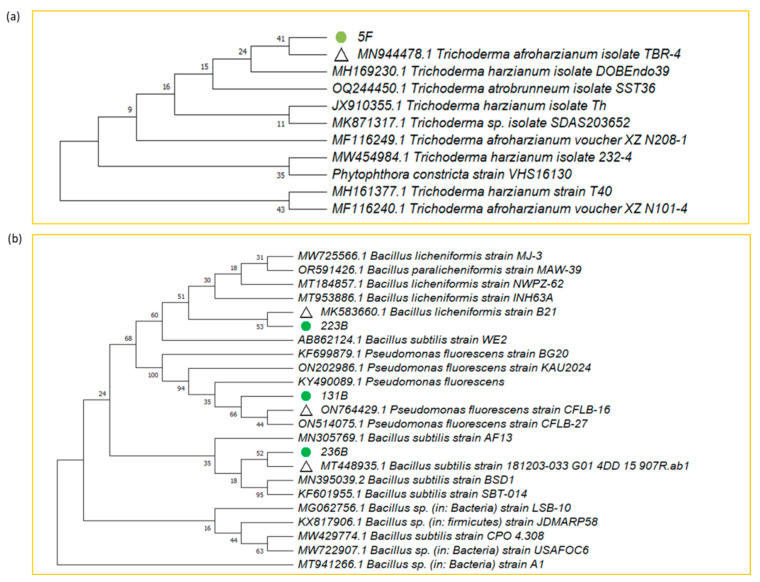
Maximum likelihood phylogenetic tree of selected fungal (**a**) and bacterial (**b**) biocontrol agents based on 16S rRNA and ITS gene sequencing (Tamura–Nei Model, 1000 Bootstrap Replications).

**Table 1 microorganisms-12-02331-t001:** Key characteristics of selected biocontrol agents.

Isolates	Siderophore	HCN	Ammonia	Chitinase	Beta-1,3, Glucanase	Cellulase	Chitosanase	IAA	P Solubilization	Zn Solubilization
1F	++	+++	++	+++	++	++	++	++	+	+
5F	+++	+++	++	+++	+++	+++	++	++	+	+
37F	++	++	++	++	+++	++	+++	++	+	+
42F	+++	++	+++	+++	++	++	++	++	+	+
131B	+++	++	+++	+++	+++	−	+	+++	++	+++
223B	+++	++	+++	++	++	−	+++	+++	+++	+++
236B	+++	++	+++	++	+++	−	++	+++	+++	+++
242B	++	+	++	+	++	−	+	+	++	++
325B	+	+	+	+	+	−	+	++	++	++
390B	++	+	++	++	+	−	++	++	+	+

where, “+” shows positive test, “+, ++ and +++” shows a lower, medium and higher ability for particular screening test, whereas “−” is for the negative test result.

**Table 2 microorganisms-12-02331-t002:** Efficacy of individual and consortium applications of biocontrol agents against *Rhizoctonia bataticola* in cluster bean pot experiment.

Treatments	PDI (Percent Disease Index)	Percent Disease Control over the Infected Control
T1: No pathogen + No biocontrol agent (mock control)	11.0 ± 2.7 ^f^	-
T2: Only *Rhizoctonia bataticola* (infected control)	94.44.0 ± 8.7 ^a^	-
T3: *Trichoderma afroharzianum* 5F + challenged with *R. bataticola*	55.55 ± 6.9 ^c^	41.2
T4: *Pseudomonas fluorescens* 131B + challenged with *R. bataticola*	61.78 ± 5.7 ^bc^	34.6
T5: *Bacillus licheniformis* 223B + challenged with *R. bataticola*	66.67 ± 4.0 ^b^	29.4
T6: *Bacillus subtilis* 236B + challenged with *R. bataticola*	61.78 ± 5.7 ^bc^	34.6
T7: 131B + 223B + 236B + challenged with *R. bataticola*	38.88 ± 6.0 ^d^	58.8
T8: 5F + 131B + 223B + 236B + challenged with *R. bataticola*	22.2 ± 2.0 ^e^	76.5

Data are the average of three replicates ± SD; Grouping information between mean values of obtained data was carried out by Fisher LSD Method and 95% confidence (*p* ≤ 0.05). Different letters point out significant differences in a column.

**Table 3 microorganisms-12-02331-t003:** Influence of biocontrol agents on biochemical characteristics and antioxidant-defense enzyme activity in cluster bean exposed to *Rhizoctonia bataticola*.

Treatments	Total Phenol (mg Gallic Acid g^−1^ F.W.)	Flavonoids (mg Quercitin g^−1^ F.W.)	Peroxidase (U min^−1^ g^−1^ F.W.)	Polyphenol Oxidase (U min^−1^ g^−1^ F.W.)	Phenylalanine Ammonia Lyase (U h^−1^ g^−1^ F.W.)	Tyrosine Ammonia Lyase (U h^−1^ g^−1^ F.W.)
T1: No pathogen + No biocontrol agent (mock control)	2.8 ± 0.2 ^d^	15.2 ± 2.1 ^d^	1.9 ± 0.2 ^c^	9.2 ± 1.3 ^c^	3.2 ± 0.6 ^e^	18.1 ± 2.2 ^e^
T2: Only *Rhizoctonia bataticola* (infected control)	3.8 ± 0.7 ^cd^	21.3 ± 3.2 ^d^	2.6 ± 0.3 ^c^	13.2 ± 1.0 ^c^	5.2 ± 0.2 ^d^	24.4 ± 2.1 ^de^
T3: *Trichoderma afroharzianum* 5F + challenged with *R. bataticola*	5.3 ± 0.6 ^b^	35.8 ± 3.4 ^c^	5.2 ± 0.8 ^b^	20.1 ± 2.1 ^b^	7.1 ± 1.0 ^bc^	39.5 ± 5.0 ^bc^
T4: *Pseudomonas fluorescens* 131B + challenged with *R. bataticola*	4.9 ± 0.5 ^bc^	30.2 ± 4.0 ^c^	4.8 ± 0.9 ^b^	21.1 ± 2.0 ^b^	6.4 ± 0.9 ^bcd^	31.2 ± 4.0 ^cd^
T5: *Bacillus licheniformis* 223B + challenged with *R. bataticola*	4.7 ± 0.7 ^bc^	32.0 ± 4.0 ^c^	4.9 ± 0.6 ^b^	21.4 ± 2.1 ^b^	6.2 ± 0.4 ^cd^	33.4 ± 5.0 ^c^
T6: *Bacillus subtilis* 236B + challenged with *R. bataticola*	4.1 ± 0.8 ^bcd^	30.3 ± 4.4 ^c^	5.0 ± 0.8 ^b^	19.8 ± 1.7 ^b^	6.1 ± 0.9 ^cd^	34.1 ± 4.1 ^c^
T7: 131B + 223B + 236B + challenged with *R. bataticola*	7.2 ± 1.3 ^a^	43.4 ± 4.1 ^b^	6.1 ± 1.2 ^b^	26.7 ± 1.3 ^a^	7.9 ± 1.3 ^ab^	45.0 ± 8.1 ^b^
T8: 5F + 131B + 223B + 236B + challenged with *R. bataticola*	7.3 ± 1.1 ^a^	51.3 ± 4.4 ^a^	7.5 ± 1.0 ^a^	28.5 ± 5.1 ^a^	8.9 ± 1.5 ^a^	56.0 ± 7.1 ^a^

Data are the average of three replicates ± SD; Grouping information between mean values of obtained data was carried out by Fisher LSD Method and 95% confidence (*p* ≤ 0.05). Different letters point out significant differences in a column.

**Table 4 microorganisms-12-02331-t004:** Impact of individual and consortium applications of biocontrol agents on cluster bean growth, biomass, yield, and yield attributes in response to *Rhizoctonia bataticola*.

Treatments	Plant Height (cm) at 70 Days After Sowing (DAS)	Fresh Weight (g) (70 DAS)	Dry Weight (g) (70 DAS)	Number of Pods Plant^−1^	Yield (g Pot^−1^)
T1: No pathogen + No biocontrol agent (mock control)	63.5 ± 4.0 ^c^	63.0 ± 4.5 ^b^	16.5 ± 2.6 ^bc^	19.0 ± 2.0 ^a^	32.0 ± 1.5 ^ab^
T2: Only *Rhizoctonia bataticola* (infected control)	51.25 ± 4.0 ^d^	35.0 ± 4.0 ^c^	9.0 ± 1.7 ^e^	3.0 ± 1.0 ^e^	1.3 ± 0.4 ^e^
T3: *Trichoderma afroharzianum* 5F + challenged with *R. bataticola*	63.0 ± 7.0 ^c^	62.3 ± 5.1 ^b^	16.5 ± 1.3 ^bc^	15.0 ± 2.0 ^bc^	24.8 ± 3.2 ^cd^
T4: *Pseudomonas fluorescens* 131B + challenged with *R. bataticola*	69.0 ± 4.5 ^abc^	60.0 ± 4.9 ^b^	15.3 ± 1.0 ^bcd^	14.0 ± 1.0 ^bcd^	21.0 ± 1.7 ^d^
T5: *Bacillus licheniformis* 223B + challenged with *R. bataticola*	67.0 ± 4.1 ^bc^	59.8 ± 4.8 ^b^	14.9 ± 1.3 ^cd^	13.0 ± 1.0 ^cd^	20.1 ± 2.2 ^d^
T6: *Bacillus subtilis* 236B + challenged with *R. bataticola*	67.3 ± 4.9 ^bc^	56.3 ± 4.1 ^b^	13.3 ± 1.4 ^d^	12.0 ± 2.0 ^d^	19.2 ± 2.5 ^d^
T7: 131B + 223B + 236B + challenged with *R. bataticola*	73.0 ± 3.6 ^ab^	72.8 ± 5.9 ^a^	18.0 ± 2.2 ^ab^	16.0 ± 1.0 ^b^	27.4 ± 5.8 ^bc^
T8: 5F + 131B + 223B + 236B + challenged with *R. bataticola*	77.0 ± 5.4 ^a^	75.8 ± 4.3 ^a^	19.5 ± 1.3 ^a^	21.0 ± 1.0 ^a^	36.1 ± 6.2 ^a^

Data are the average of three replicates ± SD; Grouping information between mean values of obtained data was carried out by Fisher LSD Method and 95% confidence (*p* ≤ 0.05). Different letters point out significant differences in a column.

## Data Availability

The original contributions presented in this study are included in the article/Appendix A. Further inquiries can be directed to the corresponding author.

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
