# Peer review of "Isolation and Characterization of Biocontrol Microbes for Development of Effective Microbial Consortia for Managing *Rhizoctonia bataticola* Root Rot of Cluster Bean Under Hot Arid Climatic Conditions"

_microorganisms, 2024, doi:10.3390/microorganisms12112331_

Round 1
Reviewer 1 Report
Comments and Suggestions for Authors
Manuscript entitled “Isolation and characterization of biocontrol microbes for development of effective microbial consortia for managing Rhizoctonia bataticola root rot of cluster bean under hot arid climatic conditions”. The manuscript develop native microbial consortia against Rhizoctonia bataticola, as well as promot plant growth. Several points need to be addressed before it can be accepted.
1. Line 71-74. Reports of the application of consortia in biocontrol should be detailed introduced.
2. Line 106-109. The composition of each medium should be described in supplementary files.
3. Line 120-121. Repeated.
4. Line 248-252. Is Rhizoctonia bataticola and biocontrol agents inoculated into the plant at the same time?
5. Line 327. No supplementary files in the submission system.
6. Fig.1. 236B was suggested to use the front of culture dish, same as other five strains.
7. Fig.2-5. Significance difference analysis was absent.
8. Table 2. Significance difference analysis was absent in the column of Percent disease control Over the infected control.
9. Line 416. Only 16S rRNA and ITS amplification could only identify strains to genus level, but not species level. Bacteria identify to species level need combined physiology, biochemistry, chemistry and molecular indexes. Fungi identify to species level need combined morphological and molecular indexes.
Author Response
Reviewer 1#
Comment 1. Manuscript entitled “Isolation and characterization of biocontrol microbes for development of effective microbial consortia for managing Rhizoctonia bataticola root rot of cluster bean under hot arid climatic conditions”. The manuscript develop native microbial consortia against Rhizoctonia bataticola, as well as promot plant growth. Several points need to be addressed before it can be accepted.
Reply: Thank you for your valuable feedback on the manuscript.
Comment 2. Line 71-74. Reports of the application of consortia in biocontrol should be detailed introduced.
Reply: It has been improved.
Comment 3. Line 106-109. The composition of each medium should be described in supplementary files.
Reply: It has been added.
Comment 4. Line 120-121. Repeated.
Reply: It has been deleted.
Comment 5. Line 248-252. Is Rhizoctonia bataticola and biocontrol agents inoculated into the plant at the same time?
Reply: No, Besides seed treatment, biocontrol agents were also applied at a rate of 10 g kg-1 of soil using a talc-based formulation at intervals of 20, 30, and 40 days after sowing (DAS). The pathogen R. bataticola was introduced through drenching treatments T2 to T8 on the 50th DAS. It is already described in text.
Comment 6. Line 327. No supplementary files in the submission system.
Reply: Supplementary files have been included.
Comment 7. Fig.1. 236B was suggested to use the front of culture dish, same as other five strains.
Reply: It has been corrected.
Comment 8. Fig.2-5. Significance difference analysis was absent.
Reply: It has been added.
Comment 9. Table 2. Significance difference analysis was absent in the column of Percent disease control Over the infected control.
Reply: The Percent Disease Control data is a comparative percentage over the infected control, serving as secondary data for this study. Therefore, a significance difference analysis is not applicable or required for this column.
Comment 10. Line 416. Only 16S rRNA and ITS amplification could only identify strains to genus level, but not species level. Bacteria identify to species level need combined physiology, biochemistry, chemistry and molecular indexes. Fungi identify to species level need combined morphological and molecular indexes.
Reply: We acknowledge the limitation that 16S rRNA and ITS amplification primarily identify strains to the genus level. To achieve species-level identification, additional data are essential. In our study, for bacterial identification, we have supplemented 16S rRNA sequencing and phylogenetic tree analysis with comprehensive data on morphology, physiology, and biochemistry. These supplementary data include detailed morphological descriptions, physiological tests (e.g., growth characteristics under various conditions), and biochemical assays (e.g., enzyme activity profiles). Similarly, for fungal identification, we have provided extensive morphological characteristics in the supplementary tables. These include descriptions of spore morphology, hyphal structures, and colony characteristics. By combining ITS sequencing with detailed morphological data, we have enhanced the resolution of fungal identification to the species level.
Reviewer 2 Report
Comments and Suggestions for Authors
Dear Author
Please find my comments below:
Page 3 line 97: How many samples were collected for each location? Season? Put it on the table. How and where are the samples kept during collection and isolation of rhizospheric and endophytic microbes?
Page 3 line 106: Cites reference on how media was prepared.
Page 3 line 113 - 117: Is all kinds of media mention above used for growing the microbes for each sample?
Page 3 line 118: What type of media used for store cultures?
Page 3 line 120-121: this is repeated?
Page 3 line 127: Be careful with writing scientific names.
Page 3 line 130: Be careful with writing scientific names.
Page 3 line 135: Be careful with writing scientific names.
Page 4 line 150: What is the purpose of growing the culture in 28C after growing in different temp? “after, all inoculated samples were incubated at 28°C for 48 hours in a shaking incubator.”
Page 4 line 155: What are the criteria for selection microbes used here?
Page 4 line 170-178: Add reference.
Page 4 line 185: Are author mean bacteria or fungi or actinomyces?
Page 6 line 247-252: Each microbe used as biocontrol should have control treatment with only the biocontrol agent itself to measure the positive or negative effect on the plant.
Page 6 line 259: Ether describes the method of preparing the inoculum or cite reference.
Page 6 line 262: What is the conditions of the greenhouse?
The biocontrol agent must be tested to host a wide range of crops because it may be pathogenic to other crops. There are papers published about the pathogenic of Trichoderma afroharzianum
Pathogenicity of Trichoderma afroharzianum in Cereal Cropsdoi: 10.3390/pathogens12070936
Author Response
Reviewer 2#
Comment 1. Page 3 line 97: How many samples were collected for each location? Season? How and where are the samples kept during collection and isolation of rhizospheric and endophytic microbes?
Reply: About 20 soil samples and 20 plant samples were collected from each location for each crop. Additionally, a total of 10 soil samples and 10 plant samples were collected from the botanical garden of ICAR-CAZRI, Jodhpur. Samples from cumin were collected during the winter season (January-February). Whereas, samples from cluster bean and moth bean were collected during the rainy season (August-September). Samples from botanical garden were collected during summer season (April-May month). Soil samples were collected in zip-type poly bags. Plant samples were placed in zip-type poly bags and kept in an ice box during transport to the laboratory. Upon arrival at the laboratory, all soil and plant samples were stored at 4°C in a refrigerator.
Comment 2. Page 3 line 106: Cites reference on how media was prepared.
Reply: It has been corrected and Supplementary table has been added.
Comment 3. Page 3 line 113 - 117: Is all kinds of media mention above used for growing the microbes for each sample?
Reply: Yes
Comment 4. Page 3 line 118: What type of media used for store cultures?
Reply: Nutrient agar slants were used for storing bacterial cultures, while potato dextrose agar slants were used for storing fungal cultures.
Comment 5. Page 3 line 120-121: this is repeated?
Reply: It has been corrected.
Comment 6. Page 3 line 127: Be careful with writing scientific names.
Reply: It has been corrected.
Comment 7. Page 3 line 130: Be careful with writing scientific names.
Reply: It has been corrected.
Comment 8. Page 3 line 135: Be careful with writing scientific names.
Reply: It has been corrected.
Comment 9. Page 4 line 150: What is the purpose of growing the culture in 28C after growing in different temp? “after, all inoculated samples were incubated at 28°C for 48 hours in a shaking incubator.”
Reply: It has been improved. Detail methodology for each parameter have been included (please see in methodology section- Assessment of selected microbial isolates for abiotic stress tolerance)
Comment 10. Page 4 line 155: What are the criteria for selection microbes used here?
Reply: Microbes were selected on the basis of in vitro antagonistic activity against R. bataticola. Based on their antagonistic activity, isolates 1F, 5F, 37F, 42F, 131B, 223B, 236B, 242B, 325B, and 390B were selected for further study (Supplementary table 2 and 3). This is mentioned in result section.
Comment 11. Page 4 line 170-178: Add reference.
Reply: It has been added.
Comment 12. Page 4 line 185: Are author mean bacteria or fungi or actinomyces?
Reply: 5F refers to fungi, while 131B, 223B, and 236B are denoted as bacteria.
Comment 13. Page 6 line 247-252: Each microbe used as biocontrol should have control treatment with only the biocontrol agent itself to measure the positive or negative effect on the plant.
Reply: Reply: In our present study, the objective was to evaluate the effect of biocontrol agents both individually and in consortia on cluster bean plants challenged with R. bataticola. To assess the impact accurately, we used two control treatments: (a) Negative control (mock control) with neither pathogen nor biocontrol agent, and (b) Positive control (infected control) where only the pathogen was present. Biocontrol-treated plants were compared with these two control treatments to determine the effectiveness of the biocontrol agents in managing the disease.
Comment 14. Page 6 line 259: Either describes the method of preparing the inoculum or cite reference.
Reply: Reference has been added.
Comment 15. Page 6 line 262: What is the conditions of the greenhouse?
Reply: Under natural environmental conditions.
Comment 16. The biocontrol agent must be tested to host a wide range of crops because it may be pathogenic to other crops. There are papers published about the pathogenic of Trichoderma afroharzianum.
Reply: We acknowledge the importance of testing the biocontrol agent on a wider range of crops to ensure it does not exhibit pathogenicity to other plant species. In future studies, we plan to expand our testing to include additional crops to further evaluate the safety and efficacy of Trichoderma afroharzianum as a biocontrol agent.
Round 2
Reviewer 1 Report
Comments and Suggestions for Authors
Accept in present form
Reviewer 2 Report
Comments and Suggestions for Authors
Dear Author
The manuscript improved and looks great. I will accept the manuscript as its.
Thanks